# Extrapolating Large Language Models to Non-English by Aligning Languages

## Abstract

Existing large language models show disparate capability across different languages, due to the imbalance in the training data. Their performances on English tasks are often stronger than on tasks of other languages. In this paper, we empower pre-trained LLMs on non-English languages by building semantic alignment across languages. We start from targeting individual languages by performing cross-lingual instruction-tuning (**CoIT**) on LLaMA, i.e. tuning it with translation task data and cross-lingual general task data to obtain cross-lingual models (x-LLaMAs), and formulate underlying scaling laws to investigate the advantages of using scalable translation data. Then we perform multilingual instruction-tuning (**MuIT**) with mixed resources to build multilingual m-LLaMA. We also illustrate how we leverage the scaling laws to optimize data allocation in a resource-constrained setting. Experiment results on cross-lingual benchmarks XQUAD and MLQA show that x-LLaMAs surpass the English instruction-tuned counterpart (Alpaca) by an average of 27.83% across six non-English languages. Evaluation results on translation dataset Flores-101 show that x-LLaMAs outperform previous LLaMA-based models by an average of 18.89%. Encouragingly, m-LLaMA achieves comparable performance to x-LLaMAs on individual languages and demonstrates the ability to follow multilingual instructions. Further analysis on response content and representation space reveals the alignment of the multilingual semantic space within the middle layers of m-LLaMA.

## 1 Introduction

The language ability of LLMs is often imbalanced across languages (Zhu et al., 2023; Yang et al., 2023; Zhang et al., 2023), because both the pre-training corpus (Blevins & Zettlemoyer, 2022) and the instruction-tuning data (Wang et al., 2023b) are English-dominated. As a result, LLMs usually perform poorly on non-English languages, especially on languages that are dissimilar to English (Bang et al., 2023; Huang et al., 2023).

There have been some attempts to enhance LLMs' non-English abilities by continued pre-training with large scale monolingual corpus (Cui et al., 2023; Yang et al., 2023). However, learning a language from monolingual data may need large scale data and computing.

In this paper, we elicit the non-English ability of pre-trained LLMs by building semantic alignment between English and non-English. To extrapolate the English ability to a particular non-English language, we propose a multi-task setting which combines translation tasks and cross-lingual general tasks during instruction-tuning. The translation tasks are used to stimulate the semantic alignment between languages, while the cross-lingual general tasks enhance the instruction-following capabilities of models (Fig 1). These cross-lingual instruction tuning (CoIT) brings out a cross-lingual model tailored to a specific non-English language.

Next, we explore to extrapolate LLM to multiple languages simultaneously through multilingual instruction-tuning (MuIT) with mixed multilingual resources (Fig 1). We consider two specific settings in our study. In the first setting, we simply combine all available resources for instruction-tuning to obtain multilingual LLM. In the second setting, we consider a practical scenario where instruction-tuning is performed under with a specific data budget. To achieve optimal data allocation, we formulate the task as non-linear programming based on our previously discovered scaling laws. The objective of this optimization is to maximize the averaged multilingual performance.

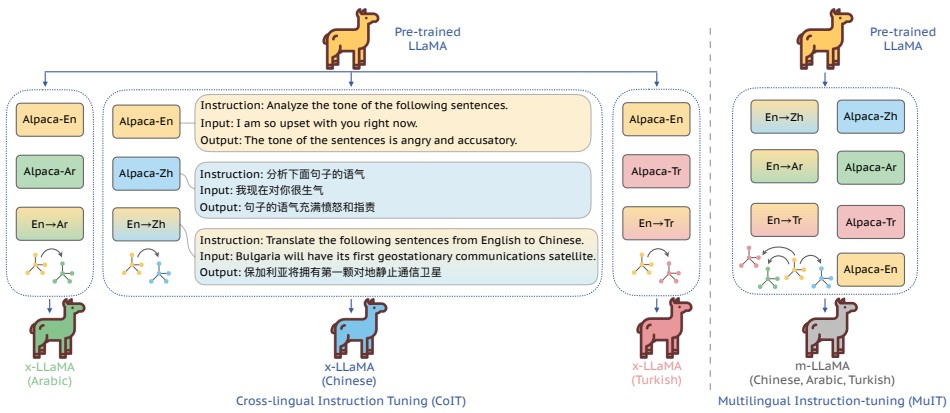

Figure 1: Illustration of cross-lingual instruction-tuning (**CoIT**) and multilingual instruction-tuning (**MuIT**). We perform cross-lingual instruction-tuning by tuning pre-trained LLM with both cross-lingual general task instruction data and translation task instruction data. We mix multilingual resources together and perform multilingual instruction-tuning to extrapolate pre-trained LLM to multiple languages simultaneously.

In the experiments, we use LLaMA-7B as the pre-trained LLM and consider six challenging target languages[1] that share little alphabet with English. For each language, a separate x-LLaMA is obtained with language-specific data. And a m-LLaMA is obtained with mixed multilingual data.

Experiment results on two cross-lingual benchmarks XQUAD and MLQA show that x-LLaMAs outperforms the model tuned with English instructions (Alpaca-7B) by an average of 27.83%. Notably, the accuracy of x-LLaMAs on non-English tasks is comparable to the performance of Alpaca-7B on English tasks. We also observe that x-LLaMAs exhibit strong translation ability without the need of massively continued pre-training. Evaluation results on multilingual translation dataset FLORES-101 show that x-LLaMAs outperforms previous LLaMA-based models by an average of 18.89% and even outperforms the supervised multilingual translation system M2M-12B (Fan et al., 2021) in half of the evaluated translation directions.

In our first setting of multilingual instruction-tuning, we discover that m-LLaMA can achieve comparable performance to strong x-LLaMAs on individual languages. Moreover, m-LLaMA is now capable of following multilingual instructions. Further analysis on response content and representation space reveals that m-LLaMA has a tendency to generate non-English response based on its English memory and multilingual semantic space becomes aligned in the middle layers of m-LLaMA, demonstrating the effectiveness of our methods. In our second setting, experiments on data allocation shows that our optimized mixture yields higher multilingual performance than a uniform mixture. This showcases a practical usage of our formulated scaling laws.

The main contribution of this paper can be summarized as:

- We explore cross-lingual instruction-tuning (CoIT) and multilingual instruction-tuning (MuIT) to elicit the non-English ability of LLMs.
- Experiment results demonstrate that our instruction-tuning methods can simultaneously boost LLM's non-English language ability, e.g., following multilingual instructions, generating multilingual response, and translation ability.
- We formulate the scaling law in cross-lingual instruction-tuning and devise a data allocation strategy based on formulated laws for resource-constrained multilingual instruction-tuning.
- We compare the scaling law of cross-lingual instruction-tuning and continued pre-training, and show that aligning language is a more efficient choice.

## 2 BACKGROUND

To unlock the potential of pre-trained LLMs, Wei et al. (2022) propose *instruction-tuning*. In this stage, LLM will be fed with instruction data $\{T, X, Y\}$, where $T$ is a task instruction that describes

---

[1]The six languages are Arabic (Ar), Greek (El), Hindi (Hi), Turkish (Tr), Vietnamese (Vi), Chinese (Zh).

the task requirement. $X$ is an optional input and $Y$ is the desired output for the given task. The objective of this optimization stage is to minimize following the negative log-likelihood.

$$\arg\min_{\theta} \frac{1}{|\mathcal{D}|} \sum_{\{T,X,Y\}\in\mathcal{D}} -\log p(Y|T,X) \tag{1}$$

where $\theta$ denotes learnable parameters of the LLM and $\mathcal{D}$ represents the instruction tuning dataset. The instruction-tuning dataset often covers diverse tasks, which is found beneficial for generalization to unseen instructions and tasks (Wei et al., 2022). However, we notice that commonly-used instruction-tuning datasets, e.g., ALPACA (Taori et al., 2023), FLAN (Longpre et al., 2023) are English-dominant, which limits LLM's potential on following non-English instructions and solving non-English tasks.

## 3 ELICITING LLM'S NON-ENGLISH ABILITY

Empowering LLM on more languages beyond English is non-trivial. Training a LLM from scratch for each non-English language is almost prohibitive due to the huge cost of data collection and computation. In this paper, we explore to elicit pre-trained LLM's non-English ability by strengthening semantic alignment between English and target languages. We begin by targeting a single language by performing cross-lingual instruction-tuning (CoIT, §3.1). To better understand the potential of aligning languages, we design a formulation to describe the scaling law in cross-lingual instruction-tuning (§3.2). In the end, we introduce multilingual instruction-tuning (MuIT), aiming at eliciting LLM's language ability on multiple non-English languages simultaneously and presents a potential usage of the scaling law in multilingual data allocation (§3.3).

### 3.1 CROSS-LINGUAL INSTRUCTION-TUNING

To elicit LLM's non-English ability, we perform cross-lingual instruction-tuning (illustrated in Fig 1) with multi-task data, including cross-lingual general task instruction data $\mathcal{D}_G$ and translation task instruction data $\mathcal{D}_T$.

**General task instruction data** $\mathcal{D}_G$    Considering that commonly-used instruction-tuning dataset is almost in English, we translate it to a foreign version with a translation engine. We then utilize both English and non-English version as cross-lingual general task instruction data. This approach aims to encourage LLM to better comprehend and follow cross-lingual instructions.

**Translation task instruction data** $\mathcal{D}_T$    Intuitively, translation data is a valuable resource for learning semantic alignment. Previous researches have also shown that LLM's translation performance can be enhanced by using expert-annotated translation data Jiao et al. (2023); Zhang et al. (2023) for instruction-tuning. Unlike them, we use publicly available parallel corpora, e.g., WIKIMATRIX (Schwenk et al., 2021), NEWSCOMMENTARY (Tiedemann, 2012), to construct translation task instruction data, making our method more reproducible, scalable and extendable to more languages. While both En-X (translating English to non-English) and X-En (translating non-English to English) translation data are beneficial for learning semantic alignment, we find that placing non-English text on the target side of translation data yields better performance improvements for LLMs on non-English tasks compared to placing it on the source side. This finding will be further demonstrated in the upcoming experiments.

### 3.2 SCALING LAW OF CROSS-LINGUAL INSTRUCTION-TUNING

We use bilingual translation performance as an indicator of semantic alignment and find that the scale of translation task instruction data has a huge impact on it. To quantify the relationship between translation performance $\mathcal{S}$ and translation data scale $\mathcal{X}$, we formulate the underlying scaling law based on following intuitions: (1) The upper bound of $\mathcal{S}$ is 100, which is the maximum score of frequently used translation quality metrics such as COMET and BLEU. (2) Translation performance tends to improve as the translation data scale increases. (3) Languages that are less similar to English

require a larger amount of translation data to establish semantic alignment compared to languages more similar to English. Consequently, we present our final formulation as follows:

$$S(\mathcal{X}) = 100 - \alpha \cdot (\gamma \cdot \mathcal{X})^{\beta} \qquad (2)$$

where $\alpha > 0$ and $\beta \in (-1, 0)$ are parameters to estimate, $\gamma \in (0, 1)$ is the language similarity[2] the target language and English. When estimating the scaling law for a specific language, we first compute $\gamma$ and then estimate $\alpha$ and $\beta$ with observed data points. In the following subsection, we will further demonstrate how these scaling laws can assit us in optimizing data allocation when constructing multilingual LLM in a resource-constrained scenario.

### 3.3 Multilingual Instruction-tuning

While cross-lingual instruction-tuning is effective, serving customized LLMs for each language can be costly, particularly as the number of languages increases. Therefore, we take a step further and investigate the possibility of extrapolating a singe pre-trained LLM to multiple non-English languages simultaneously.

To achieve this goal, we perform multilingual instruction-tuning with a combination of multilingual resources. This includes general task instruction data in multiple languages and translation task instruction data from multiple directions. By leveraging these resources, the instruction-tuned LLM can establish alignment between English and multiple languages, enabling it to comprehend and follow multilingual instructions.

As for data mixture, we consider two settings. In the first setting, we straightforwardly combine all available resources for instruction-tuning. But a potential drawback of this approach is that instruction-tuning LLM with large-scale multilingual data may take huge computational cost.

Therefore we also consider a practical scenario, where the available instruction data is constrained by a specific data budget. For instance, the total amount of utilized parallel data is a fixed number. To achieve the optimal data combination in this scenario, we propose to formulate data allocation as a non-linear programming problem. The objective of this programming problem is to maximize the averaged multilingual performance:

$$\max \frac{1}{n} \sum_{i=1}^{n} S(\mathcal{X}_i), \quad \text{s.t.} \quad \sum_{i=1}^{n} \mathcal{X}_i = C, \quad \text{where} \quad 0 \leq \mathcal{X}_i \leq \mathcal{X}_i^{max}, \quad i = 1, 2, 3 \cdots, n$$

There are two constraints in this formulation: (i) *data budget*, the total amount of translation task instruction data is limited by a fixed budget $C$. (ii) *data availability*, the maximum number of available translation data for language $i$ is $\mathcal{X}_i^{max}$.

## 4 Experiment Setting

**Pre-trained LLM**   We take LLaMA-7B as the pre-trained LLM, which is trained on trillions of tokens (mainly in English) and found to be competitive with state-of-the-art LLMs (Touvron et al., 2023). We construct x-LLaMAs for six challenging target languages: Arabic (Ar), Greek (El), Hindi (Hi), Turkish (Tr), Vietnamese (Vi) and Chinese (Zh), which share little alphabet with English.

**Baseline LLMs**   For comparison, we include several models that are built by instruction tuning on LLaMA: Alpaca-7B (Taori et al., 2023), which is tuned with English instructions; Parrot-7B (Jiao et al., 2023), which is tuned with human annotated translation data; Bayling-7B (Zhang et al., 2023), which is tuned with human interactive translations and English instruction data. We also present results from Chinese-Alpaca-7B (Cui et al., 2023) and Bigtrans-13B (Yang et al., 2023) for reference. Both these two models extend the vocabulary of LLaMA and use a large scale monolingual data for continued pre-training.

---

[2]We calculate language similarity following the approach of Pan et al. (2021) using multi-way translation data.

**Instruction tuning details**  For translation task instruction data, we use publicly available parallel corpora, WIKIMATRIX[3] (Schwenk et al., 2021) and NEWSCOMMENTARY[4] (Tiedemann, 2012). These corpora are more accessible and scalable compared to high-cost expert-annotated data (Jiao et al., 2023; Zhang et al., 2023). The statistics of two datasets are presented in Table 1. For multilingual general task instruction data, we incorporate ALPACA dataset (Taori et al., 2023), which consists of 52k English questions and corresponding response, and we obtain its foreign version with in-house translation engine. We use *stanford_alpaca*[5] as the code base. More training details are provided in Appendix A.

| Parallel Corpora | Ar | El | Hi | Tr | Vi | Zh |
|---|---|---|---|---|---|---|
| WIKIMATRIX | 999.8k | 620.8k | 231.5k | 477.7k | 1073.8k | 786.5k |
| NEWSCOMMENTARY | 97.4k | - | 2.8k | - | - | 126.0k |
| TOTAL | 1097.2k | 620.8k | 234.3k | 477.7k | 1073.8k | 912.5k |

Table 1: Statistics of parallel corpora. In our experiments, we use above two open-source parallel corpora to construct translation task instruction data.

**Evaluation Dataset**  To evaluate LLM's performance on non-English languages, we use two benchmark cross-lingual datasets, XQUAD (Artetxe et al., 2020) and MLQA (Lewis et al., 2020), which requires the model to reason over the given context and answer the given question. In addition, we create a new multilingual evaluation set MI-EVAL (introduced in Appendix B) to assess the capability of LLM in following multilingual instructions. These multilingual multi-way test sets allow us to compare language ability across languages. To evaluate LLM's translation ability, we use multilingual translation dataset FLORES-101 (Goyal et al., 2022). Details of the prompts used for all these tasks are provided in Appendix C.

**Evaluation Metrics**  On XQUAD, MLQA and MI-EVAL, we follow Liu et al. (2023) and Wang et al. (2023a) to use ChatGPT for generation quality evaluation. On XQUAD, MLQA, we also report exact-matching results in Appendix D. For translation tasks, we use COMET (Rei et al., 2020), BLEURT (Sellam et al., 2020) and sentence-piece BLEU (Papineni et al., 2002) as metrics[6]. More evaluation details can be referred to Appendix D.

## 5   MAIN RESULTS

### 5.1   RESULTS ON CROSS-LINGUAL INSTRUCTION-TUNING

**x-LLaMA achieves great improvement on non-English QA tasks**  Table 2 presents experimental results for non-English question answering tasks. We can see that Alpaca-7B performs poorly on non-English, although it achieves 95% answer accuracy on corresponding English questions. Notably, x-LLaMA outperforms its counterpart (Alpaca-7B) by an average of 27.83% across six non-English languages. More importantly, x-LLaMA's answer accuracy on non-English tasks is approaching Alpaca-7B's answer accuracy on English tasks. This indicates that cross-lingual instruction-tuning is an effective way to elicit LLM's non-English ability.

Table 2 also reports multilingual performance of other two representative LLaMA-based models. Both Chinese-Alpaca-7B (trained on a large-scale Chinese corpus) and Bayling-7B (trained on chinese-annotated interactive translation data) shows impressive performance on Chinese task. But they does not perform well on five other languages. Since their used training data can not be easily obtained, it is hard to extend their training frameworks to cover more languages.

---

[3]https://opus.nlpl.eu/News-Commentary.php
[4]https://github.com/facebookresearch/LASER/tree/main/tasks/WikiMatrix
[5]https://github.com/tatsu-lab/stanford_alpaca
[6]Specifically, we report COMET score computed by *wmt22-comet-da* model and report BLEURT score computed by *BLEURT-20* model.

| System | XQUAD | | | | | | MLQA | | | |
|---|---|---|---|---|---|---|---|---|---|---|
| | Ar | El | Hi | Tr | Vi | Zh | Ar | Hi | Vi | Zh |
| Alpaca-7B | 0.36 | 0.64 | 0.41 | 0.51 | 0.45 | 0.59 | 0.39 | 0.36 | 0.50 | 0.67 |
| Chinese-Alpaca-7B | 0.34 | 0.22 | 0.20 | 0.22 | 0.35 | **0.91** | 0.38 | 0.22 | 0.34 | **0.97** |
| Bayling-7B | 0.44 | 0.48 | 0.53 | 0.55 | 0.58 | 0.88 | 0.43 | 0.55 | 0.54 | 0.89 |
| x-LLaMA-7B | **0.82** | **0.75** | **0.81** | **0.64** | **0.81** | 0.80 | **0.85** | **0.76** | **0.93** | 0.81 |

Table 2: Evaluation results on XQUAD and MLQA dataset. The highest answer accuracy in each column is indicated in bold. For reference, Alpaca-7B's answer accuracy on corresponding English task is 0.95 on XQUAD and 0.96 on MLQA.

**x-LLaMA shows impressive translation performance** Figure 2 and Figure 9 (Appendix E) present the performance of x-LLaMA and baseline systems in translating between English and non-English, which serves as crucial evidence for language alignment. Compared with other LLaMA-based LLMs, x-LLaMA exhibits higher translation performance in all evaluated directions. Notably, x-LLaMA even outperforms strong supervised baseline M2M-12B (Fan et al., 2021) on four En-X directions (translating English to non-English) and two X-En directions (translating non-English to English), and is approaching strong commercial translation engines, ChatGPT and Google Translate.

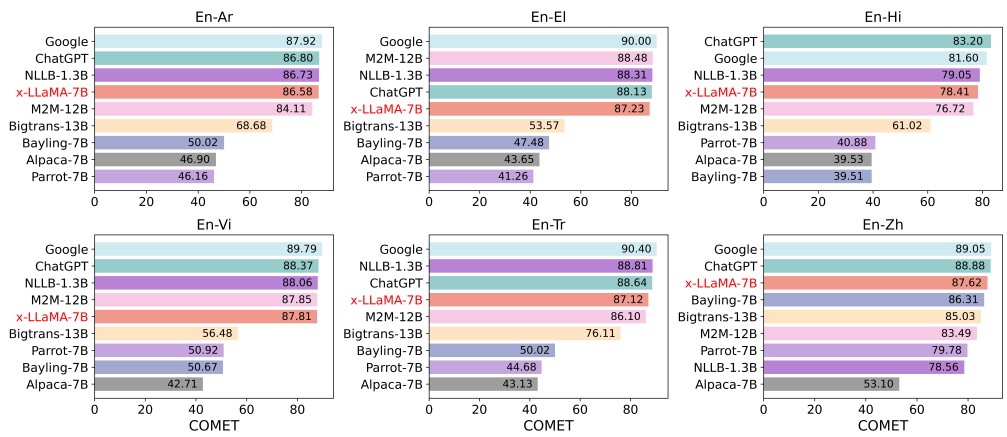

Figure 2: Performance of different systems in translating English to non-English.

**Scaling laws of cross-lingual instruction-tuning** Now, we investigate x-LLaMA's translation performance under varying translation data scales and present the advantages of using scalable translation task instruction data for language alignment. As illustrated in Figure 3, adding translation data is always beneficial for strengthening semantic alignment. Encouragingly, our designed formulation (represented by the dotted line) effectively captures the trend and provide quantified relationship between translation performance and translation data scale. In subsequent experiments, we will demonstrate the practical applications of these formulated scaling laws, e.g., optimizing data allocation, analyzing learning efficiency.

## 5.2 RESULTS ON MULTILINGUAL INSTRUCTION-TUNING

**m-LLaMA achieves comparable performance to x-LLaMAs on individual languages.** In our first setting, we combine all available resources to construct m-LLaMA. We then compare the performance of m-LLaMA with that of x-LLaMAs customized for individual languages. Figure 4 shows that m-LLaMA can achieve comparable performance to x-LLaMAs on non-English QA tasks and multilingual translation tasks. This indicates the feasibility of extrapolating pre-trained English LLaMA to multiple non-English languages simultaneously.

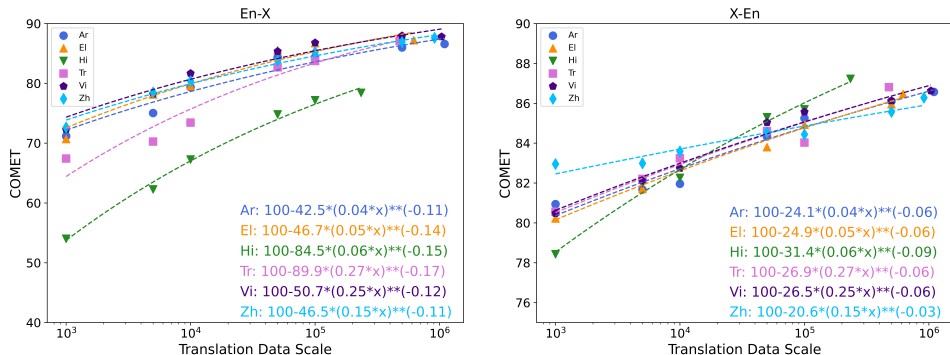

Figure 3: The relationship between translation data scale and translation performance (COMET). Our designed formulation (the dotted line) fits well with the trend and we list estimated scaling laws in the figure.

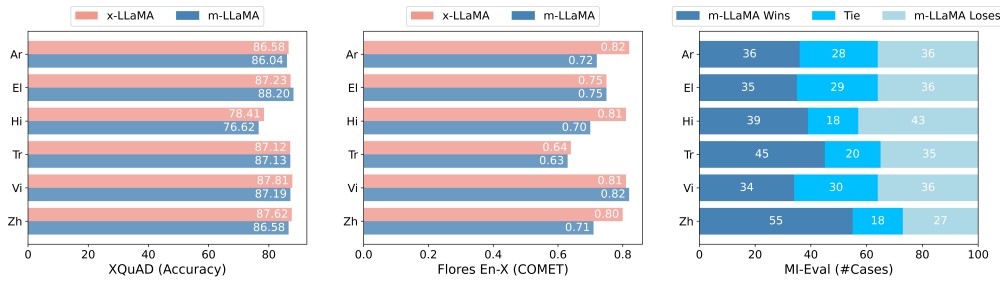

Figure 4: Performance comparison between m-LLaMA and x-LLaMAs on multilingual question answering (left), multilingual translation (middle) and following multilingual instruction (right).

**m-LLaMA is able to handle multilingual instructions according to its English memory.** More importantly, incorporating multiple languages into a single m-LLaMA enables it to follow multilingual instructions. Evaluation results on MI-EVAL (Fig. 4) show that m-LLaMA can achieve comparable response quality to x-LLaMAs when provided with instructions in different languages.

Additionally, we find that our instruction-tuning approach has minor impact on LLM's English proficiency[7] but makes m-LLaMA have a tendency to generate response to non-English instructions with its English memory. Table 3 show two representative cases where m-LLaMA produces similar response when given instructions in different languages. This phenomenon suggests that English and non-English becomes aligned within LLM after our instruction-tuning.

**Visualization results show that multilingual semantic space becomes aligned in the middle layers of m-LLaMA.** For comprehensive analysis, we investigate the representation space of m-LLaMA and Alpaca-7B. Specifically, we use them to encode multilingual multi-way data from FLORES-101 dataset and compare encoded representations across different layers. Figure 5 displays visualization results. For Alpaca-7B, representations of different languages always stay apart from bottom layers to top layers. In contrast, we observe representation overlap in m-LLaMA, especially in the middle layers, which offers another evidence that multilingual instruction-tuning encourages language alignment.

**Our formulated scaling laws can be used to optimize data allocation for data-constrained multilingual instruction-tuning.** In our second setting, we study data-constrained multilingual instruction-tuning and explore the usage of formulated scaling laws. We compare our devised allo-

---

[7]We draw this conclusion by comparing the answer quality of m-LLaMA and Alpaca-7B on English test set of MI-EVAL. Generally, their answer quality is close. On 24% of test cases, m-LLaMA wins. On 32% of test cases, m-LLaMA loses. On the rest of test cases, the two models tie.

| m-LLaMA (Chinese instruction) | m-LLaMA (English instruction) |
|---|---|
| [Input]
选择三个形容词来描述一条街道 | [Input]
Choose three adjectives to describe a street |
| [Output]
繁华、充满活力和美丽。(Bustling, vibrant and beautiful.) | [Output]
Busy, vibrant, and lively. |
| [Input]
民主社会努力维护哪些价值观? | [Input]
What values does a democratic society strive to uphold? |
| [Output]
民主社会努力维护的价值观包括个人自由、平等、言论和宗教自由、人身安全、公正法律和人类尊严。民主社会还促进尊重多样性和包容性,以及言论自由和集体决策的权利。(The values that democratic societies strive to uphold include individual liberty, equality, freedom of speech and religion, security of the person, just laws, and human dignity. Democratic societies also promote respect for diversity and inclusion, as well as the rights to freedom of expression and collective decision-making.) | [Output]
A democratic society strives to uphold values such as freedom, equality, justice, and human rights. It is committed to upholding the right of all citizens to participate in decision-making and to be treated with respect and dignity. It also seeks to promote freedom of speech and thought, as well as the rule of law. |

Table 3: Two representative cases where m-LLaMA makes similar response when given instructions in different languages. The gray text in the bracket denotes the English meaning of Chinese response.

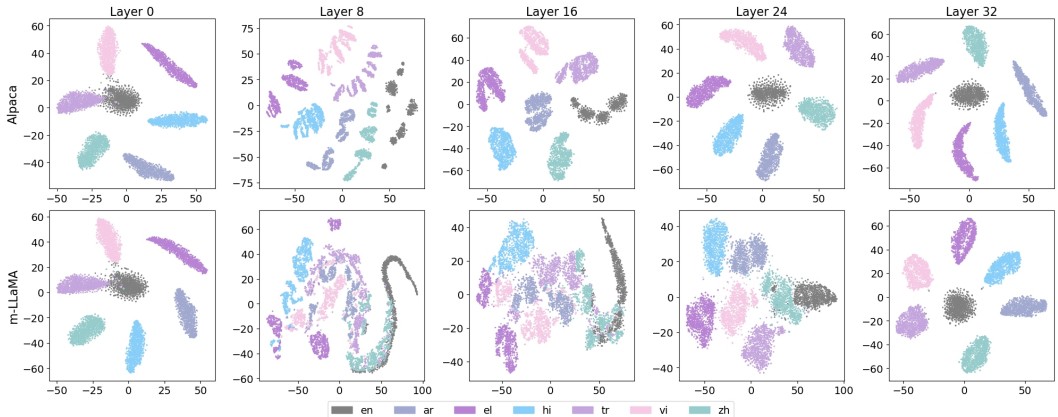

Figure 5: Visualization analysis on the representation space of m-LLaMA and Alpaca-7B. For Alpaca-7B, representations of different languages always stay apart from bottom layers to top layers. In contrast, we observe representation overlap in m-LLaMA, especially in middle layers.

cation approach with uniform data allocation in Table 4. The experiment results is mixed. When the data budget is low, e.g., 300k, the gap between different data allocation strategies is minor. When the data budget reaches 1.2M, our optimized allocation achieves significantly higher averaged multilingual translation performance than uniform allocation in all three metrics, which meets our optimization objective.

| Budget | Data Allocation | | | | | | FLORES-101 | | |
|---|---|---|---|---|---|---|---|---|---|
| | Ar | El | Hi | Tr | Vi | Zh | COMET | BLEURT | BLEU |
| 300k | 50,000 | 50,000 | 50,000 | 50,000 | 50,000 | 50,000 | 82.13 | 66.36 | 30.05 |
| | 41,842 | 44,953 | 73,002 | 59,652 | 40,731 | 39,816 | 82.38 (+0.25) | 66.75 (+0.39) | 30.21 (+0.16) |
| 1.2M | 200,000 | 200,000 | 200,000 | 200,000 | 200,000 | 200,000 | 84.22 | 69.73 | 33.81 |
| | 183,539 | 189,556 | 234,233 | 242,263 | 175,985 | 174,422 | 84.70*(+0.48) | 70.42*(+0.69) | 34.40*(+0.59) |

Table 4: Comparison results between our optimized allocation and uniform allocation. The number in the bracket denotes the performance gap between the two data allocation strategies. The annotation "*" indicates that the improvement is significant ($p < 0.1$).

## 6    ANALYSIS AND DISCUSSION

**Ablation study**    We conduct experiments with different combinations of instruction data for ablation study (Table 6). Instruction tuning LLaMA-7B with Chinese Alpaca data is better than English Alpaca data on the Chinese task. Jointly using two versions of Alpaca data brings further improvement. Interestingly, using translation task instruction data alone can reach moderate answer accuracy. And we find that putting Chinese on the target side of translation data is more useful for boosting LLaMA's non-English ability. Jointly using cross-lingual general task instruction data and En-Zh translation task instruction data reaches the highest accuracy.

**Using translation data is far more efficient than monolingual data for building semantic alignment.**    Using monolingual corpus of target language for continued pre-training is another way to help LLM to understand non-English and improve translation performance (Yang et al., 2023). For comparison, we use Chinese monolingual corpus MC4 (Xue et al., 2021) for continued pre-training and use cross-lingual general task data for instruction-tuning. Figure 7 compares scaling laws of two approach. We observe that using parallel data is far more efficient than using monolingual data for accomplishing semantic alignment.

| Alpaca-En | Alpaca-Zh | En-Zh | Zh-En | XQUAD |
|:---:|:---:|:---:|:---:|:---:|
| ✓ | | | | 0.59 |
| | ✓ | | | 0.66 |
| | | ✓ | | 0.60 |
| ✓ | ✓ | | | 0.75 |
| ✓ | ✓ | ✓ | | 0.80 |
| ✓ | ✓ | | ✓ | 0.60 |

Figure 6:    Ablation study of cross-lingual instuction-tuning on Chinese XQUAD. "Alpaca-En" and "Alpaca-Zh" denotes original and foreign version Alpaca data. "En-Zh" and "Zh-En" denotes the direction of translation data.

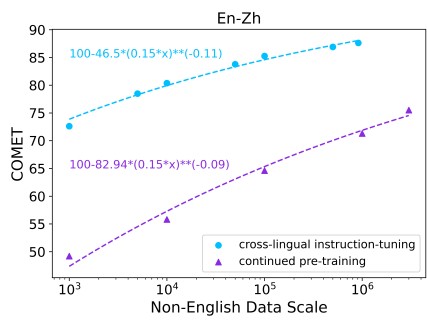

Figure 7:    Comparison between scaling laws of cross-lingual instruction-tuning and continued pre-training.

**Discussion on extending vocabulary for non-English.**    Unlike previous work (Cui et al., 2023; Yang et al., 2023), we do not extend vocabulary for target non-English languages. The effect is dual. Our approach does not require a large-scale non-English corpus to learn embedding of extended tokens. On the other hand, since LLaMA usually tokenizes non-English tokens to bytes, our model is slower in encoding and decoding non-English sequence than those models equipped with extended vocabulary. We leave the exploration on vocabulary manipulation as our future work.

## 7    CONCLUSION

In this paper, we focus on extrapolating pre-trained large language models to non-English by building semantic alignment across languages. Specifically, we explore two approach: cross-lingual instruction-tuning (CoIT) and multilingual instruction-tuning (MuIT). Experiment results show that our cross-lingual models, x-LLaMAs, achieve great improvements on non-English, e.g., outperforming its English counterpart (Alpaca-7B) by 27.83% on question answering tasks and by 18.89% on translation tasks. After training on mixed multilingual resources, our m-LLaMA model can achieve comparable performance to strong x-LLaMAs on individual languages and is capable of following multilingual instructions. Further analysis of response consistency and representation space reveals that multilingual semantic space becomes aligned in the middle layers of m-LLaMA. In the setting of resource-constrained multilingual instruction-tuning, we show the usage of formulated scaling laws to achieve optimal data allocation. Overall, our approach and findings illuminate the potential for developing more potent LLMs for non-English languages.

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

## A   DETAILS OF OUR INSTRUCTION-TUNING

For each experiment, we instruction-tune LLaMA's full parameters for 3 epoch on 8×A100. The learning rate is set as 2e-5 and batch size is set as 128. For training acceleration, we adopt FSDP training strategy (Zhao et al., 2023).

## B   DETAILS OF OUR CONSTRUCTED MI-EVAL DATASET

We follow the fashion of "self-instruct" (Wang et al., 2022) and generate new English instructions with Alpaca dataset as seed. Then we translate these instructions to six non-English languages with strong multilingual machine translation system NLLB Costa-jussà et al. (2022) and obtain the multilingual multi-way evaluation set MI-EVAL.

| Task | Dataset | Prompt |
|---|---|---|
| Question Answering | XQUAD, MLQA (English) | Answer the final question with following context.
Context: <Context>
Question: <Question>
Answer: |
| Question Answering | XQUAD, MLQA (Arabic) | الرجاء الإجابة على الأسئلة بناءً على الفقرات التالية
فقرة: <Context>
سؤال: <Q>
إجابة: |
| Question Answering | XQUAD, MLQA (Greek) | Απαντήστε στις ερωτήσεις με βάση τις παρακάτω παραγράφους
παράγραφος: <Context>
ερώτηση: <Question>
Απάντηση: |
| Question Answering | XQUAD, MLQA (Hindi) | कृपया निम्नलिखित पैराग्राफ के अनुसार प्रश्नों के उत्तर दें
अनुच्छेद: <Context>
सवाल: <Question>
उत्तर: |
| Question Answering | XQUAD, MLQA (Turkish) | Lütfen soruları aşağıdaki paragraflara göre cevaplayınız
paragraf: <Context>
soru: <Question>
Cevap: |
| Question Answering | XQUAD, MLQA (Vietnamese) | Hãy trả lời các câu hỏi theo đoạn văn sau
đoạn văn: <Context>
câu hỏi: <Question>
Trả lời: |
| Question Answering | XQUAD, MLQA (Chinese) | 请根据以下段落，回答问题
段落: <Context>
问题: <Question>
答案: |
| Machine Translation | Flores-101 | Translate the following sentences from <SRC> to <TGT>. |

Figure 8: Our used prompts for downstream tasks. In the prompt for question answering tasks, <Context> and <Question> denote the placeholder for the given context and question. In the prompt for machine translation tasks, <SRC> and <TGT> denote English name of source and target language.

## C    OUR USED PROMPTS FOR DOWNSTREAM TASKS

We report all our used prompts in Figure 8. For question answering tasks, i.e., XQUAD and MLQA, we apply language-specific prompt when evaluate LLM's performance on the target language. Table 5 lists two cases for better illustration. For machine translation tasks, i.e. FLORES-101, we use English instruction for multilingual translation in our experiments.

---

**[Input (English prompt)]**
Answer the final question with following context
Context: The Broncos defeated the Pittsburgh Steelers in the divisional round, 23−16, by scoring 11 points in the final three minutes of the game. They then beat the defending Super Bowl XLIX champion New England Patriots in the AFC Championship Game, 20−18, by intercepting a pass on New England's 2-point conversion attempt with 17 seconds left on the clock. Despite Manning's problems with interceptions during the season, he didn't throw any in their two playoff games.
Question: Who did the Broncos defeat in the AFC Championship game?
Answer:

**[Output]**
New England Patriots

---

**[Input (Chinese prompt)]**
请根据以下段落，回答问题
段落：野马队在分区轮以23−16击败了匹兹堡钢人队，在比赛的最后三分钟拿下11分。然后他们在美式足球联合会(AFC)锦标赛上以20−18击败了第49届超级碗卫冕冠军新英格兰爱国者队，在比赛还剩17秒时拦截了新英格兰队的两分转换传球。尽管曼宁在本赛季的拦截上有问题，但他在两场季后赛中未投任何球。
问题：野马队在AFC锦标赛中打败了谁？
答案：

**[Output]**
野马队在AFC锦标赛中打败了新英格兰爱国者队。

---

Table 5: Cases of using our prompt for handling QA tasks.

## D    DETAILS OF CONDUCTING EVALUATION WITH CHATGPT

In this paper, we use ChatGPT to automatically evaluate the quality of question answering and instruction following. The evaluation prompts are reported below. Considering the API cost of evaluating with ChatGPT, we use the first one hundred questions in XQUAD and MLQA as representatives for experiments. Table 6 also reports exact-matching results on full test set. However, we notice two limits of exact-matching evaluation: (1) it does not penalty answer that heavily copies the given context. (2) it does not favor answer that is correct but different from the reference answer.

---

**Evaluating Answer Quality with ChatGPT**

**Prompt:**
You will be given a context followed by question. You will then be given one potential answer to the question. Your task is to tell if the answer is correct. Please make sure you read and understand these instructions carefully. Please keep this document open while reviewing, and refer to it as needed.

Evaluation Criteria: Correctness (YES or NO): Is the answer correct? YES means the answer provides an accurate and valid response that aligns with the facts, logic, and requirements of the question. The answer should be in the same language as the context. NO means otherwise.
Context: <Context & Question>
Answer: <Answer>

Evaluation Form (YES or NO):

---

> **Comparing Response Quality with ChatGPT**
>
> **Prompt:**
> We would like to request your feedback on the performance of two AI assistants in response to the user question displayed above. Please rate the helpfulness, relevance, accuracy, level of details of their responses.
>
> Each assistant receives an overall score on a scale of 1 to 10, where a higher score indicates better overall performance. Please first provide a comprehensive explanation of your evaluation, avoiding any potential bias and ensuring that the order in which the responses were presented does not affect your judgment. Then, output two lines indicating the scores for Assistant 1 and 2, respectively.
>
> Output with the following format:
> Evaluation evidence: <Explanation>
> Score of the Assistant 1: <Score>
> Score of the Assistant 2: <Score>

| System | XQUAD | | | | | | MLQA | | | |
|---|---|---|---|---|---|---|---|---|---|---|
| | Ar | El | Hi | Tr | Vi | Zh | Ar | Hi | Vi | Zh |
| Alpaca-7B | 0.16 | 0.53 | 0.21 | 0.49 | 0.51 | 0.32 | 0.10 | 0.10 | 0.34 | 0.31 |
| Chinese-Alpaca-7B | 0.29 | 0.11 | 0.07 | 0.32 | 0.18 | **0.72** | 0.20 | 0.09 | 0.07 | 0.58 |
| Bayling-7B | 0.35 | 0.43 | 0.46 | 0.51 | 0.53 | 0.62 | 0.25 | **0.31** | 0.35 | **0.59** |
| x-LLaMA-7B | **0.49** | **0.56** | **0.59** | **0.61** | **0.54** | 0.53 | **0.27** | **0.31** | **0.47** | 0.48 |

Table 6: Evaluation results on XQUAD and MLQA dataset (measured by exact matching). The bold text denotes the highest answer accuracy along the column. For reference, Alpaca-7B's answer accuracy on corresponding English task is 0.88 on XQUAD and 0.66 on MLQA.

## E TRANSLATION PERFORMANCE ON REVERSE TRANSLATE DIRECTIONS

Due to the page limit of main text, we report translation performance of different systems on translating non-English to English here (Fig. 9). The findings are similar to those in §5.1.

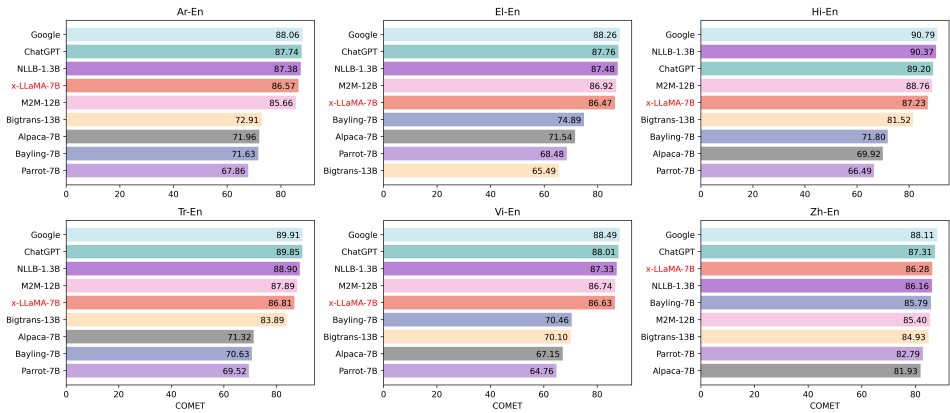

Figure 9: Performance of different systems on translating non-English to English.

