# OpenReview forum: "Extrapolating Large Language Models to Non-English by Aligning Languages"
_ICLR.cc/2024/Conference — ICLR 2024 Conference Withdrawn Submission_

### Official Review · Reviewer_bT9A · 2023-10-30

**Soundness:** 3 good
**Presentation:** 3 good
**Contribution:** 3 good
**Rating:** 5
**Confidence:** 4

**Summary:**

This paper propose to enhance the low-resource language ability of  LLM with a multi-task tuning framework which combines the translation tasks and cross-lingual general task. Specifically two tuning method is introduced: cross-lingual instruction tuning and multilingual instruction tuning.
Experiment shows that the proposed x-LLaMA models achieves significant improvement on non-English languages,
and the language alignment is improvement measured by the improvement in machine translation results.

**Strengths:**

The paper is clearly written and easy to understand.
The proposed CoIT and MuIT method is solid in improving the cross-lingual ability of pre-trained language models.
And the formulated scaling laws to optimize the data allocation of data-constrained MuIT is novel to me.
And I like the experiment design and analysis: 1. Extensive experiment on 6 languages and 3 kinds of dataset to show the improvement of cross-lingual ability.  2. Sophisticated experiment on machine translation task, with various evaluation metrics.

**Weaknesses:**

I have two concerns about this paper:
1. There is not enough related work comparison: in Table 2, the proposed work is only compared to other LLaMA-based model tuned without multi-lingual dataset, so the improvement sees very significant, but I think it also should be compared with some other cross-lingual instruction tuning methods like "Few-shot Learning with Multilingual Generative Language Models", with the same multi-lingual training data to really reflect the merits of this work.
2. The author argues that the CoIT could improve the semantic alignment.  I think only using translation performance to quantify the improvement in alignment is not enough, since the evaluation metrics like Comet and BLEU doesn't always correlate with better semantic quality. So maybe add the distance between the representations of positive/negative samples is worth trying.

**Questions:**

See weakness for detail:
1. Is there a better related work comparison to justify this work?
2. Could the improvement in semantic alignment could be better measure?

---

> ### Author Response · Authors · 2023-11-21
> **Response to reviewer-bT9A**
>
> W1: The proposed work is only compared to other LLaMA-based model
>
> The comparison with other llama-based models aims to avoid the influence of different base models on the comparison results. To attain a more meaningful comparison, it might be more insightful to compare different data combinations (as shown in the table below). Such comparisons highlight the significance of our method: compared to using only English Alpaca data, both the Chinese Alpaca data and translation data contribute to the improvement. The combination of these datasets achieves the highest performance.
>
> | Data  | XQUAD (exact match) | MLQA (exact match) | mLAMA (exact match) | XLSum (Rouge-1) |
> | --- | --- | --- | --- | --- |
> | Alpaca-En | 31.8 | 26.7 | 5.3 | 9.0 |
> | Alpaca-En + En-Zh | 34.3 | 38.0 | 5.8 | 27.1 |
> | Alpaca-En + Alpaca-Zh | 51.7 | 48.0 | 21.9 | 25.5 |
> | Alpaca-En + Alpaca-Zh + En-Zh | 54.9 | 51.8 | 30.4 | 28.3 |
>
> W2: Quantify the improvement in alignment by measuring the distance between the representations of positive/negative samples
>
> Thank you for your insightful comments. Following your advice, we use English and Chinese as an example and measure the cosine distance between positive pairs and negative pairs across different layers (0, 8, 16, 24, 32). By 'positive pair', we refer to an English sentence and its corresponding Chinese translation. By 'negative pair', we mean an English sentence and a randomly sampled Chinese sentence. As shown below, in the representation of x-LLaMA, positive pairs stay closer than in Alpaca, especially in the latter half of the model layers.
>
> |  |  | 0 | 8 | 16 | 24 | 32 |
> | --- | --- | --- | --- | --- | --- | --- |
> | x-LLaMA | cosine distance between positive pair | 0.15  | 0.98  | 0.95  | 0.88  | 0.13  |
> |  | cosine distance between negative pair | 0.10  | 0.98  | 0.92  | 0.75  | 0.01  |
> | Alpaca | cosine distance between positive pair | 0.15  | 0.98  | 0.91  | 0.71  | 0.01  |
> |  | cosine distance between negative pair | 0.11  | 0.98  | 0.89  | 0.63  | -0.08  |

---

### Official Review · Reviewer_yxQQ · 2023-10-31

**Soundness:** 3 good
**Presentation:** 3 good
**Contribution:** 2 fair
**Rating:** 5
**Confidence:** 4

**Summary:**

This paper proposes a strategy to fine-tune a cross-lingual LLM (x-LLaMa) using cross-lingual instruction datasets and create multilingual LLMs (m-LLaMa) based on mixtures of training resources, including parallel corpora. The objective is to enhance the semantic alignment across multiple languages, thus improving the performance of a pre-trained LLM in non-English languages. Experimental results show that x-LLaMa and m-LLaMa achieve better performance on non-English tasks and translation tasks than previous LLaMa-based models. Ablation studies were conducted to demonstrate the effectiveness of the proposed strategy, including the use of different scales of data for tuning the models. Both the results and analyses reveal interesting findings for LLMs.

**Strengths:**

1.	This paper systematically presents a study demonstrating the use of multi-lingual instruction datasets along with multi-lingual parallel datasets to fine-tune pre-trained LLMs for non-English languages, achieving comparable or even better performance than previous models.
2.	The findings and conclusions drawn from this study provide valuable insights to the community, indicating that using translation data can enhance language abilities beyond English.
3.	Overall, the paper is well-organized and well-written, making it easy to follow.

**Weaknesses:**

1.	One of the contributions of the paper is the proposal of the scaling law to quantify the relationship between translation performance and the size of the training data. However, there is a lack of insightful discussions and analyses regarding this aspect. For instance, Eq. (2) considers the language similarity between English and the target language, but unfortunately, the computational details are not provided or explained, which makes it difficult to replicate the work.
2.	Another concern is that the authors only verify their proposal using the LLaMa-7B model. It remains unclear if the conclusion still holds for other LLMs, such as Bloom.

**Questions:**

1.	How is the similarity of language computed, and how are the values of α and β estimated for the scaling law?
2.	In Table 4, it is unclear how the optimal allocation of data is obtained. It would be helpful to provide further elaboration on this.

**Details Of Ethics Concerns:**

NIL

---

> ### Author Response · Authors · 2023-11-21
> **Response to reviewer-yxQQ**
>
> W1: There is a lack of insightful discussions and analyses regarding scaling law
>
> Thank you for your advice. We designed the current scaling law based on three motivations, which are discussed in section 3.2. In our preliminary experiments, we also explored different formula design options, and eventually settled on the best solution. In the next version, we can expand the discussion on this topic, providing more insightful analyses.
>
> We can also explain the computation details regarding language similarity here. Our computation process aligns closely with that of Pan et al. Initially, we encode multi-lingual parallel data using LLM, and average the representations from the final layer to derive sentence vectors. To illustrate with the calculation of English-Chinese language similarity, we use the representations of English sentences to retrieve Chinese sentences. A retrieval score can be determined based on the ranking of the target sentence in the retrieval results. By averaging all retrieval scores, we can obtain the language similarity between English and Chinese. The higher the similarity, the closer the internal representations of parallel sentences within the LLM.
>
> W2: It remains unclear if the conclusion still holds for other LLMs.
>
> To address your concern, we implemented our methods on Pythia-6.9B [1], another English-dominant LLM. Below, we present experiment results on Chinese tasks. The conclusion mirrors our previous findings: both Chinese Alpaca data and translation data prove to be beneficial for enhancing the Chinese ability of the LLM.
>
> | System  | XQUAD (exact match) | MLQA (exact match) | mLAMA  (exact match) | XLSum (Rouge-1) | Flores En-Zh (COMET) |
> | --- | --- | --- | --- | --- | --- |
> | Pythia + [ Alpaca-En] | 22.0 | 18.0 | 0.5 | 13.1 | 62.5 |
> | Pythia + [ Alpaca-En + Alpaca-Zh] | 40.2 | 39.1 | 15.7 | 37.4 | 70.3 |
> | Pythia + [ Alpaca-En + Alpaca-Zh + En-Zh ] | 44.3 | 42.7 | 19.6 | 28.6 | 85.5 |
>
> [1] Biderman et al. Pythia: A suite for analyzing large language models across training and scaling.
>
> Q1: How is the similarity of language computed, and how are the values of α and β estimated for the scaling law?
>
> For the process of computing language similarity, please refer to our response to W1. As for the process of computing α and β, we can explain it here: given a set of {scale, performance} data points, we use a Python package to automatically estimate α and β. The values derived in this manner allow the scaling law curve to best fit the known data points.
>
> Q2: Further elaboration on how the optimal allocation of data is obtained.
>
> We obtain the optimal allocation by solving a non-linear programming problem, as introduced in Section 3.3. The optimization solution is achieved using the Python package scipy.optimize.
>
> The underlying insight is: during this process, the data is preferentially allocated to the language with greater marginal benefit (the gradient of the scaling law, i.e., the performance improvement that can be brought by adding a unit of data). When each unit of data is allocated to a language to maximize its benefits, the maximum total benefit is achieved (which is reflected in the average multi-language translation performance).

---

### Official Review · Reviewer_55oU · 2023-11-03

**Soundness:** 2 fair
**Presentation:** 2 fair
**Contribution:** 2 fair
**Rating:** 3
**Confidence:** 4

**Summary:**

The paper effectively achieves cross-lingual transfer by employing fine-tuning techniques on LLMs using a parallel corpus. The authors introduce two distinct forms of training data, specifically a parallel corpus and a translation corpus, which are employed to fine-tune the model. Subsequently, the model is evaluated on the XQuAD and MLQA datasets. The experimental results demonstrate the effectiveness of their approach in tackling these tasks.

**Strengths:**

1. The collected parallel corpus in different languages is invaluable and can greatly contribute to future research.

2. The performance of the tested tasks and languages has shown remarkable improvement.

**Weaknesses:**

1. The method employed lacks novelty, as it primarily involves the collection of a parallel corpus and subsequent fine-tuning of the model.

2. The method lacks testing on high-resource languages closely related to English, such as French or German, as well as on low-resource languages like Thai or Swahili.

3. The method has not been sufficiently tested on multilingual NLP tasks, including reasoning tasks such as MGSM and XCOPA, as well as NLG tasks like XLSum.

4. The tested results on Flores are not convincing, especially when considering that the model has primarily been fine-tuned for translation tasks.  Additionally, for the translation training data, please refer to the following question 3.

**Questions:**

1. Is the translator engine effectively aligned with human-annotated data in terms of quality? Furthermore, have any techniques been implemented to ensure the removal or filtration of irrelevant or meaningless data from the training process?

2. In the CoIT example depicted in Figure 1, it is notable that the Chinese parallel corpus still includes English phrases such as 'Instruction:' and 'Input:'. It would be beneficial to analyze to assess the significance of these prompts and their impact on the overall performance.

3. Regarding the two types of training data, given the availability of a parallel corpus, it is worth investigating whether the translation corpus truly has a significant impact. Conducting an ablation analysis could provide valuable insights in this regard. Furthermore, expanding on this point would contribute to a more comprehensive understanding of the overall training process.

---

> ### Author Response · Authors · 2023-11-21
> **Response to reviewer-55oU**
>
> W1: The employed method lacks novelty
>
> The reviewer might overlook the following technical novelty in our paper.
> 1. We introduce multi-task instruction-tuning as a strategy to extrapolate LLM's english ability to non-English.
> 2. We quantify the scaling law in cross-lingual instruction-tuning and use these laws to optimize data allocation in a resource constrained setting.
>
> W2: The method lacks testing on high-resource languages closely related to English, as well as on low-resource languages.
>
> In our experiments, we do have results for low-resource languages, such as Hindi. We acknowledge that we did not consider languages closely related to English, and we will include relevant results in our next version.
>
> W3: The method has not been sufficiently tested on multilingual NLP tasks, including reasoning tasks such as MGSM and XCOPA, as well as NLG tasks like XLSum.
>
> Thank you for your suggestion. We report results on additional tasks below, including knowledge assessment (mLAMA), summarization (XLSum), and reasoning (mGSM). The experimental phenomena on mLAMA and XLSum are similar to previous experiments. mGSM is an exception here, as all models perform poorly on this task. We believe this dataset poses a significant challenge, and further exploration may be needed to improve model reasoning capabilities and facilitate the transfer of reasoning abilities across different languages.
>
> | Data  | mLAMA (exact match) | XLSum (Rouge-1) | mGSM (exact-match) |
> | --- | --- | --- | --- |
> | Alpaca-En | 5.3 | 9.0 | 4.4 |
> | Alpaca-En + En-Zh | 5.8 | 27.1 | 1.2 |
> | Alpaca-En + Alpaca-Zh | 21.9 | 25.5 | 2.4 |
> | Alpaca-En + Alpaca-Zh + En-Zh | 30.4 | 28.3 | 3.6 |
>
> Q1: Is the translator engine effectively aligned with human-annotated data in terms of quality?
>
> We understand your concern. The translator we utilize is a high-quality commercial translation engine. To check the quality of the data, we sampled and manually checked a portion of it, and found that the translated Alpaca data is accurate and reliable.
>
> Q2: It would be beneficial to analyze to assess the significance of these prompts and their impact on the overall performance.
>
> The depiction in Figure 1 is merely for illustration. In reality, during the instruction-tuning process, we consistently use the Alpaca template [1] to wrap the instruction, input and output - a common practice in this field. We adopted this method in all our experiments, thus holding this experimental variable constant. Therefore, we believe it won't influence the conclusions drawn from our experiments.
> [1] Taori et al. Stanford Alpaca: An Instruction-following LLaMA model.
>
> Q3: Conducting ablation analysis would contribute to a more comprehensive understanding of the overall training process.
>
> Following your advice, we conducted experiments to demonstrate the value of both Chinese Alpaca data (Alpaca-Zh) and English-Chinese translation data (En-Zh). Compared to using only English Alpaca data, both the Chinese Alpaca data and the translation data contribute to improvement. The combination of these datasets achieves the highest performance.
>
> | Data  | XQUAD (exact match) | MLQA (exact match) | mLAMA (exact match) | XLSum (Rouge-1) |
> | --- | --- | --- | --- | --- |
> | Alpaca-En | 31.8 | 26.7 | 5.3 | 9.0 |
> | Alpaca-En + En-Zh | 34.3 | 38.0 | 5.8 | 27.1 |
> | Alpaca-En + Alpaca-Zh | 51.7 | 48.0 | 21.9 | 25.5 |
> | Alpaca-En + Alpaca-Zh + En-Zh | 54.9 | 51.8 | 30.4 | 28.3 |

---

### Official Review · Reviewer_Hw1M · 2023-11-05

**Soundness:** 2 fair
**Presentation:** 2 fair
**Contribution:** 1 poor
**Rating:** 3
**Confidence:** 5

**Summary:**

In this paper, the authors propose a method for multilingual instruction tuning. The approach relies on jointly tuning parallel corpora and translated instruction-tuning data. The model is compared with other public multilingual instruction-tuned models. However, due to many lacunae in the experimental setup (described later) the paper cannot establish that the proposed approach is better than previously proposed approaches. The paper also introduces a multilingual benchmark (MI-EVAL) which is an automatically translated version of ALPACA (the automatic translation is a major limitation of this dataset).

**Strengths:**

* The paper studies joint finetuning on parallel corpora and translated instruction data. Though this setting has been studied in previous work (Parrot), this paper makes an attempt to study various possibilities.
* Ablation studies comparing monolingual pre-training, translation data instruction tuning, and multilingual task instruction tuning. These establish that tuning on translation data and translated instruction data is useful. However, the analysis still has some unanswered questions which are mentioned in the Weaknesses section.

**Weaknesses:**

* The proposed model is compared primarily with Bayling/Chinese Alpaca. Chinese Alpaca is trained only on Chinese. Hence, its results in Chinese are better than the proposed model, whereas it underperforms on other languages. On the other hand, Bayling is finetuned on 4 European languages, while this paper evaluates other languages. This evaluation setup is not a fair comparison to establish that the proposed approach is better than the previous work. This does not clearly answer any research question. Why not compare with truly multilingual models like Bactrian-X (or finetune alternative models to study various experimental configurations as described next).
* To understand how/if the proposed approach is indeed better, the following additional ablations would help. Is finetuning on English IFT data (Aplaca-En) plus translation data (En-Zh) sufficient to achieve crosslingual transfer? Is cross-lingual instruction tuning necessary?
* To establish the generalization of these results, the ablation results should be reported on all languages considered, not just Chinese.
* The research questions the paper seeks to answer are not presented with clarity. How do the ablations answer all those questions?  In its current form, the paper proposed a model but does not clearly articulate how this model improves over current work. The experimental setup also doesn’t lend itself to answering the research questions clearly as mentioned earlier.
* With only limited exposure to foreign languages (via instruction tuning), how can the model achieve enough proficiency to generate fluent target languages?
* It is known that a high-quality parallel corpus is needed for finetuning. It is also known that WikiMatrix is noisy. Given that there are many high-quality corpora available, why should they not be used for translation data finetuning?
* The MI-Eval has been created via automatic translation. How good is the translation quality, and what is its impact on evaluation? Results on an error-prone translated dataset with no quality evaluation are not sufficient to measure model capabilities. Moreover, ChatGPT has been used for evaluation. How good is ChatGPT for the evaluation of open-ended tasks for non-English languages? There is no evidence of the efficacy of this evaluation methodology.

**Questions:**

* The paper mentions that the performance on English tasks is not impacted, and an example is provided. It would be good to report performance on English QA benchmarks before and after finetuning with cross-lingual data.
* MI-EVAL: The English side looks like a replication of ALPACA. Why was ALPACA not directly used prior to translation?

---

> ### Author Response · Authors · 2023-11-21
> **Response to reviewer-Hw1M**
>
> W1: The current evaluation setup is not fair enough, preventing the production of effective conclusions.
>
> Thank you for your comments. We acknowledge that comparing our models with Chinese-Alpaca and Bayling may cause confusion as they are not directly comparable. In our next version, we will exclude their results and instead provide a more meaningful comparison.
>
> W2: Ablation study is needed to demonstrate the value of added data.
>
> Following your advice, we conduct experiments to demonstrate the value of both Chinese Alpaca data (Alpaca-Zh) and English-Chinese translation data (En-Zh). Compared to using only English Alpaca data, both the Chinese Alpaca data and the translation data contribute to improvement. The combination of these datasets achieves the highest performance.
> |Instruction Data|XQUAD (exact match)|MLQA (exact match)|mLAMA (exact match)|XLSum (Rouge-1)|
> |-|-|-|-|-|
> |Alpaca-En|31.8|26.7|5.3|9.0|
> |Alpaca-En+En-Zh|34.3|38.0|5.8|27.1|
> |Alpaca-En+Alpaca-Zh|51.7|48.0|21.9|25.5|
> |Alpaca-En+Alpaca-Zh+En-Zh|54.9|51.8|30.4|28.3|
>
> W3: the ablation results should be reported on all considered languages.
>
> Following your advice, we update ablation results on all considered languages on MLQA dataset (results on El, Ru, Tr are not rpeorted because MLQA does not support these languages). The conclusion is consistent: both the Alpaca data in target language and translation data contribute to improvement and the combination of these data achieves the highest performance.
>
> |Instruction-tuning Data|Ar|Hi|Vi|Zh|
> |-|-|-|-|-|
> |Alpaca-En|16.1|13.7|34.1|26.7|
> |Alpaca-En+En-Zh|33.6|35.1|42.2|38.0|
> |Alpaca-En+Alpaca-Zh|33.1|35.1|50.1|48.0|
> |Alpaca-En+Alpaca-Zh+En-Zh|37.0|42.3|50.8|51.8|
>
> W4: The research questions the paper seeks to answer are not presented with clarity.
>
> Sorry for the unclear presentation. We hope the updated results included in this response can better clarify the contributions of our paper.
>
> W5: how can the model achieve enough proficiency to generate fluent target languages with only limited exposure to foreign languages?
>
> While utilizing a large amount of non-English data is indeed the most direct method to enhance non-English capabilities, it imposes significant demands on data scale. Our objective is to transfer English capabilities to non-English with minimal data. This approach effectively leverages the model's English capabilities and presents a more data-efficient choice.
>
> W6: Given that there are many high-quality corpora available, why should they not be used for translation data finetuning?
>
> Indeed, there are many high-quality translation datasets available for experiment, but it is not feasible for us to try them all. To our knowledge, NewsCommentary and WikiMatrix are two representative high-quality translation datasets. Through our experiments on these two datasets, we aim to demonstrate that it is possible to transfer English capabilities to non-English by aligning languages.
> In line with your suggestion, we also present results using other translation datasets. We used the NIST English-Chinese training set and report the results below. As the results indicate, utilizing translation data from different sources does not seem to significantly impact the final results.
> |Data|XQUAD (exact match)|MLQA (exact match)|mLAMA (exact match)|XLSum (Rouge-1)|
> |-|-|-|-|-|
> |Alpaca-En|31.8|26.7|5.3|9.0|
> |Alpaca-En+Alpaca-Zh+En-Zh|54.9|51.8|30.4|28.3|
> |Alpaca-En+Alpaca-Zh+En-Zh (NIST) |52.0|51.3|25.3|29.5|
>
> W7: Concern regarding the creation and evaluation of the MI-Eval dataset.
>
> The original goal of creating the MI-Eval dataset was to evaluate a model's capability to follow multilingual instructions. However, we also acknowledge the difficulty in automatically evaluating the quality of an LLM's response. Using chatGPT for evaluation is a common approach, but it poses issues related to reliability, interpretability, and reproducibility. Therefore, we consider removing this part in the next version.
>
> Q1: It would be good to report performance on English QA benchmarks before and after finetuning.
>
> Following your advice, we report the performance of x-LLaMA on the English XQUAD below. Compared to the English Alpaca model, the x-LLaMA model that has been fine-tuned on different languages does not indeed show much degradation in performance on English tasks.
>
> |System|XQUAD-En (exact match)|
> |-|-|
> |Alpaca-7B|74.6|
> |x-LLaMA (Ar)|67.6|
> |x-LLaMA (El)|73.7|
> |x-LLaMA (Hi)|74.5|
> |x-LLaMA (Tr)|72.9|
> |x-LLaMA (Vi)|70.6|
> |x-LLaMA (Zh)|72.3|
>
> Q2: Difference between MI-Eval and Alpaca dataset
>
> We created the MI-Eval for evaluation instead of using Alpaca for two reasons: 1. Alpaca has been used for training, and using training data for testing may not reflect the true generalization ability of the model; 2. Alpaca is an English dataset, making it unsuitable for evaluating non-English capabilities.